# Improving Usability in Mobile Apps for Residential Energy Management: A Hybrid Approach Using Fuzzy Logic

Ivonne Nuñez [ID], Elia Esther Cano [ID], Edmanuel Cruz [ID], Dimas Concepción [ID], Nila Navarro and Carlos Rovetto *[ID]

School of Computer Systems Engineering, Universidad Tecnológica de Panamá, Panamá 0819-07289, Panama; ivonne.nunez@utp.ac.pa (I.N.); elia.cano@utp.ac.pa (E.E.C.); edmanuel.cruz@utp.ac.pa (E.C.); dimas.concepcion@utp.ac.pa (D.C.); nila.navarro@utp.ac.pa (N.N.)
* Correspondence: carlos.rovetto@utp.ac.pa

**Abstract:** This paper presents a study that evaluates the usability and user experience of a mobile application interface for residential energy management, adopting a hybrid approach that integrates quantitative and qualitative methods within a user-centered design framework. For the evaluation, metrics and tools such as the User Experience Questionnaire Short (UEQ-S) and the System Usability Scale (SUS) were used, in addition to the implementation of a fuzzy logic model to interpret and contrast the data obtained through these metrics, allowing a more accurate assessment of usability and user experience, reflecting the variability and trends in the responses. Three aspects evaluated stand out: satisfaction with the interface, ease of use, and efficiency. These are fundamental to understanding how users perceive the system. The results indicate a high likelihood of user recommendation of the system and a high overall quality of user experience. This study significantly contributes to mobile application usability, especially in residential energy management, offering valuable insights for designing more intuitive and effective user interfaces on mobile devices.

**Keywords:** UI/UX; fuzzy logic; UEQ-S; SUS; usability metrics; energy management; mobile interface

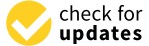



## 1. Introduction

The growing adoption of mobile applications and the increase in the number of users using them has made crucial aspects such as usability and accessibility critical to ensure that a design meets users' needs. A product needs to grab the user's attention with an engaging and enriching experience and make it easy for them to learn how to use it intuitively. The product should enable users to complete tasks quickly and efficiently while maintaining control over each stage of interaction. In this sense, the user-centered design approach emerges as a critical strategy for creating highly usable software applications. This approach is based on fundamental metrics proposed by Nielsen Norman and Krug [1,2], structured in a four-phase process: identification of usability needs and criteria, collection of crucial interface requirements, interface design and prototyping, and usability testing [3,4].

In parallel, usability stands out as a central aspect of software quality according to ISO 9241-11 [5,6], which emphasizes effectiveness, efficiency, and user satisfaction when performing specific tasks, seeking to reduce errors caused by users, and allowing them to perform tasks more efficiently and effectively, thus increasing their satisfaction and improving their overall experience with the application or system with which they interact. This practice is infused in the design and implementation stages, following clear design guidelines and using detailed evaluations during the analysis phase, which generally includes the participation of end users and experts [7–9].

User experience (UX) research plays a crucial role in digital development by thoroughly analyzing end-user behaviors and preferences [10]. The UX methodology, which uses direct observation and surveys, ensures that design decisions are well informed. The user interface (UI) links the human experience and the technical capabilities of the software,

translating needs and behaviors into tangible elements of interaction. The symbiotic relationship between UX and UI ensures that products meet users' functional demands and expectations and provide comprehensive, intuitive, and satisfying experiences [11–14]. This approach is supported by several studies [11,12,15,16] that demonstrate the direct correlation between the success of an application, specific aspects of UX/UI, and metrics such as user retention, speed of learning, ease of use, conversion rate, user satisfaction, accessibility, performance and load time, as well as direct user feedback, and highlight the importance of practical design and constant monitoring through metrics to ensure continued user adoption and satisfaction [17,18].

In this context, this paper evaluates the usability and user experience of a mobile application interface for residential energy management, adopting a hybrid approach that integrates quantitative and qualitative methods within a user-centered design framework. For the evaluation, metrics, and tools such as the User Experience Questionnaire Short (UEQ-S) and the System Usability Scale (SUS) will be employed, in addition to the implementation of a fuzzy logic model to interpret the data obtained through these metrics, allowing a more accurate assessment of usability and user experience, reflecting the variability and trends in the responses. The user-centric mobile interface design, oriented towards usability and accessibility, is expected to generate higher user satisfaction in residential energy management. This improved user satisfaction will translate into greater efficiency and effectiveness in performing energy management tasks, ultimately resulting in a more satisfying overall experience for users. The combination of these methods will enrich the research results, providing practical recommendations for improving the mobile interface based on the findings obtained from the usability evaluations and tests.

Therefore, in developing this research, we seek to answer the following hypotheses or key research questions in the design of the mobile interface. First, whether there is a positive relationship between interface design and user satisfaction in managing residential energy consumption through a mobile application. Second, whether certain design elements, such as usability and accessibility, have a significant impact on user experience, and finally, whether usability metrics such as efficiency, effectiveness, and overall user satisfaction are positively correlated in the use of the mobile application for residential energy management.

The rest of this paper is organized as follows: Section 2 presents the background of the research; Section 3 presents the materials and methods; Section 4 presents the experimentation carried out in this study; Section 5 presents the fuzzy model; Section 6 presents the results and discussions; and finally, Section 7 concludes the paper with future research perspectives.

## 2. Background

### 2.1. Related Work

In the work of Escanillan-Galera et al. [11], they present the design and development of EnerTrApp. This prototype mobile web application allows consumers to monitor the energy consumption of household appliances using their smartphones. The focus is to evaluate this application's user interface through usability testing to measure its effectiveness, efficiency, and level of user satisfaction. The results of the data analysis indicate that the EnerTrApp user interface is highly effective, evidenced by the fact that all participants completed all the proposed tasks. The application was also found to be efficient regarding both time and relative efficiency. In addition, statistical data showed that the evaluators were satisfied with the user interface of the mobile web application.

The research by Chanchí-Golondrino et al. [19] explicitly proposes developing a fuzzy logic-based tool to calculate the criticality percentage of the identified problems more efficiently, using severity and frequency values. This tool uses membership functions and inference rules defined in the Fuzzy Control Language (FCL). The work validates this tool through its application in a heuristic evaluation of the Sevenet Database Management System of Industria Licorera del Cauca, demonstrating its usefulness in determining the criticality levels of the problems identified during the inspection.

On the other hand, Berendes et al. [20] introduce a novel method called Energy System Modeling Usability Testing (ESMUT), accompanied by a detailed guide and a specific usability questionnaire (ESMUQ) based on the Post-Study System Usability Questionnaire and adapted for quantitative usability testing in open-source energy system modeling frameworks. The case study results revealed that, in general, participants were satisfied with the frameworks tested. However, input data handling and error messages were identified as frequent problems. These findings suggest that the usability of the frameworks studied needs further improvement. This study also acknowledges limitations, such as the exclusive perspective of developers and the small number of participants, indicating the need for further research to evaluate the usability of open-source power system modeling frameworks comprehensively.

*2.2. Theorical Context*

Measuring usability is not a straightforward process because it depends on multiple factors; therefore, measurement frameworks that define specific attributes of software applications and their associated metrics are used to evaluate it objectively. These attributes, considered as the characteristics or properties of an application, are essential to understanding and quantifying usability, and given their abstract nature, they cannot be measured directly; instead, metrics, numerical or nominal values derived from observable data, are assigned. These metrics must possess adequate mathematical properties and intuitively reflect the characteristics they represent, varying in correlation with the presence of positive or negative features, making it crucial to validate these metrics in various contexts before using them in decision-making. Although there are no consolidated standards for validating them, this article provides a valuable approach to this end, offering a significant contribution to the field of usability evaluation in mobile applications [21–23].

On the other hand, Nielsen's usability principles emphasize that the relevance of these attributes varies according to context and users, forming an essential framework in the design of interactive products and systems. These principles focus on the ease of learning and using the interface, as well as the ability of users to perform tasks efficiently and effectively. Usability attributes are classified into two groups: objectively quantifiable, such as efficiency and errors made, and subjectively quantifiable, such as user satisfaction, as assessed by user feedback, which is intrinsically related to the user's perception of usability [24–26].

These attributes and associated metrics provide a complete picture of the user's experience with the application, facilitating a more accurate and helpful usability assessment [27,28]. It is essential to note that, depending on the nature of the application to be evaluated, different usability measurement attributes are considered relevant; in this case, special attention will be paid to the mobile application environment, as shown in Table 1. The importance of this distinction lies in recognizing that, although certain aspects can be objectively quantified, the interpretation of the results may have subjective nuances, thus underlining the relevance of considering both factors in usability evaluations [15,29].

**Table 1.** Attributes and metrics applied to mobile environments.

| Attributes | Concept | Metrics |
|---|---|---|
| Effectiveness | Related to the accuracy and completeness with which users achieve specific objectives through the application. Key indicators include the quality of the solution and the rate of errors committed. | Tasks solved in a limited time<br>Percentage of tasks completed on the first attempt<br>Number of functions learned |
| Efficiency | Represents the relationship between effectiveness and the effort or resources invested to achieve the objectives. Relevant indicators are task completion time and learning time. Greater efficiency is achieved with less effort or resources. | Time taken to complete a task<br>Number of taps per task<br>Time spent on each screen<br>Relative efficiency compared to an expert user<br>Productive time |

**Table 1.** *Cont.*

| Attributes | Concept | Metrics |
|---|---|---|
| Satisfaction | Reflects the degree to which the user experiences satisfaction and positive attitudes when using the application to meet specific objectives. Being a subjective attribute, it is evaluated using attitude rating scales. | Difficulty level<br>Likes or dislikes<br>Preference |
| Ease of Learning | The user's ability to achieve goals in their first interaction with the application. | Time used to complete a task the first time<br>Learning curve |
| Errors | The occurrence and severity of errors made by users. It is essential to minimize errors and provide effective recovery mechanisms. | Number of errors |
| Content | It involves the layout and format of the information presented to the user. | Number of words per screen<br>Number of screens |
| Portability | The ability of the application to be moved between different platforms or environments. | Configuration level |
| Context | Refers to the factors and variables of the environment in which the application is used. | Degree of connectivity<br>Location<br>Device characteristics |
| Security | Evaluates the application's ability to handle risks and protect stored data. | User control<br>Several security rules |

## 3. Materials and Methods

### 3.1. Methodology

The MPIu + a model (Usability and Accessibility Engineering Process Model) has been used as a framework for integrating quantitative and qualitative methods within user-centered design. This approach, depicted in Figure 1, is designed to place the user at the center of the development process, with the primary goal of improving usability and user experience, as well as ensuring full accessibility of the final product. This model is structured in a series of modules or stages that define the phases of development and location of knowledge activities, facilitating user participation in all stages of the process, from requirements analysis to post-prototyping evaluation [30,31].

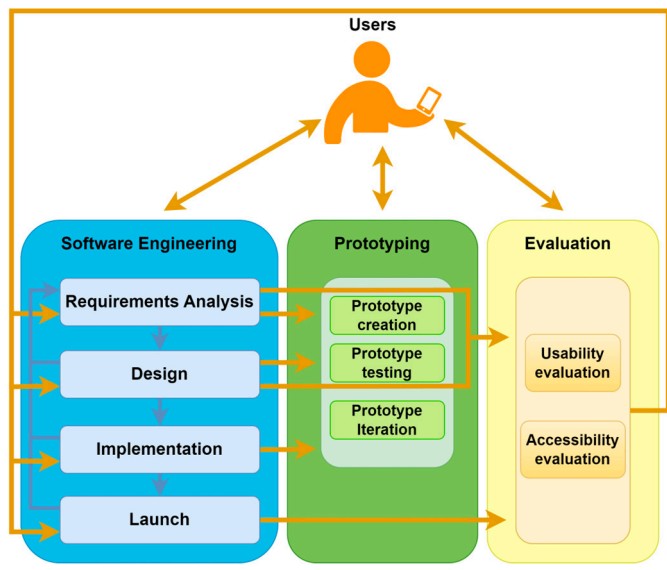

**Figure 1.** The model used in the methodology.

In the initial qualitative research phase, observations and socio-demographic surveys are conducted to gain an in-depth understanding of user needs, expectations, and preferences. These qualitative findings serve as the basis for informing the initial design of the mobile interface, ensuring that it is aligned with the expectations and requirements of the users identified at this stage.

Subsequently, quantitative methods, such as questionnaires and usability testing, are employed to collect numerical data on user effectiveness, efficiency, and satisfaction with the designed mobile interface. These quantitative data provide an objective and generalizable perspective of the interface's performance, which complements the qualitative understanding obtained earlier and allows for a more complete assessment of the user experience. The integration of these qualitative and quantitative approaches provides a more complete and balanced view of the user experience, facilitating ongoing iterations in interface design to optimize usability, accessibility, and satisfaction. In addition, this methodology not only focuses on evaluating the mobile interface itself, but also seeks to determine whether the resulting system is an effective tool and whether users are likely to use it to reduce energy consumption in their homes.

To ensure methodological robustness and validity of the results, careful selection of participants and the application of appropriate statistical analysis tools are required. Additional attention to methodological controls will strengthen the validity and applicability of the research results, thus providing a solid basis for decision-making and implementation of improvements to the mobile energy management interface.

### *3.2. Instruments*

### 3.2.1. User Experience Questionnaire Short (UEQ-S)

The UEQ-S questionnaire is an abbreviated version of the User Experience Questionnaire (UEQ) that evaluates the UX of interactive products. This questionnaire allows for quick and easy evaluation by users, using a 7-point Likert scale, with scores ranging from −3 to +3 to reflect agreement or disagreement with positive or negative terms, providing an overall measure of UX based on the average of these eight items [32,33].

The UEQ-S classifies user experience qualities into two categories, pragmatic and hedonic, divided into six scales: attractiveness, clarity, efficiency, reliability, motivation, and novelty. Items 1 to 4 in the UEQ-S evaluate pragmatic aspects, while items 5 to 8 focus on hedonic quality, as shown in Table 2 [34]. UEQ-S is based on the idea that UX can be measured considering usability and user experience goals. Efficacy, insight, and reliability are task-related aspects of user experience (usability goals). Stimulation and novelty are non-task-related aspects of the user experience (user experience objectives). Attractiveness is a task and non-task dimension [35].

**Table 2.** Items and qualities of the short version UEQ-S.

| | | | | | | | | | | |
|---|---|---|---|---|---|---|---|---|---|---|
| **Pragmatic Quality** | obstructive | O | O | O | O | O | O | O | supportive | 1 |
| | complicated | O | O | O | O | O | O | O | easy | 2 |
| | inefficient | O | O | O | O | O | O | O | efficient | 3 |
| | confusing | O | O | O | O | O | O | O | clear | 4 |
| **Hedonic Quality** | boring | O | O | O | O | O | O | O | exciting | 5 |
| | not interesting | O | O | O | O | O | O | O | interesting | 6 |
| | conventional | O | O | O | O | O | O | O | inventive | 7 |
| | usual | O | O | O | O | O | O | O | Leading edge | 8 |

### 3.2.2. System Usability Scale (SUS)

The System Usability Scale (SUS) is a tool created by John Brooke in 1986 and recognized for evaluating the usability of various products and services, including hardware, software, mobile devices, websites, and applications, to understand the problems users face while using the interface. It consists of a ten-question questionnaire (see Table 3),

with answers rated on a scale of 1 to 5, ranging from "Strongly Disagree" to "Strongly Agree" [36,37].

**Table 3.** Items System Usability Scale (SUS).

| (+) Positive | | | (−) Negative | |
|---|---|---|---|---|
| 5 | 4 | 3 | 2 | 1 |
| Strongly Agree | Somewhat Agree | Neither Agree nor Disagree | Somewhat Disagree | Strongly Disagree |
| System Usability Scale (SUS) | | | | |
| Q1: I think I would like to use this application frequently. | | | | |
| Q2: I find this application unnecessarily complex. | | | | |
| Q3: I think the application is easy to use. | | | | |
| Q4: I think I would need support to use the application. | | | | |
| Q5: I find the various functions of the application well integrated. | | | | |
| Q6: I have found too much inconsistency in this application. | | | | |
| Q7: I think most people would learn to use the application quickly. | | | | |
| Q8: I have found the app quite cumbersome to use. | | | | |
| Q9: I feel very confident using the app. | | | | |
| Q10: I would need to learn many things before operating the app. | | | | |

To obtain the SUS results, the averaged results obtained from the user questionnaires are added, considering the following: the odd questions (1, 3, 5, 7, and 9) will take the value assigned by the user, and 1 will be subtracted from it. The even questions (2, 4, 6, 8, 10) will be five minus the value assigned by the interviewees. Once the final number is obtained, it is multiplied by 2.5 [38,39].

The median score on the scale is 68. If the score is below 68, there are likely severe problems with usability that need to be identified and addressed. A score above 68 is considered positive but denotes some usability deficiency [40,41]. To complement the quantitative analysis provided by the questionnaire, we supplemented it with two qualitative questions to learn a little more about the users' opinions: What do you think is the best aspect of this interface and why? What do you think needs to be improved and why?

### 3.2.3. Fuzzy Logic

This is a mathematical tool that allows reasoning with imprecise or vague information, using partial truth values instead of absolute ones. This logic is based on degrees of membership, which vary between 0 and 1, partially allowing an element to belong to multiple sets [42]. Linguistic variables are defined in fuzzy sets, expressed in natural language, and describe varying degrees of membership. This tool fills a gap compared to other approaches, underlining its importance in providing mathematical results free of subjectivities. In contrast to more conventional methodologies, fuzzy logic stands out for its flexibility in dealing with uncertainty, making it a valuable tool for reasoning and decision-making in complex situations [43].

Fuzzy sets can be defined with Equation (1):

$$A = \{(x, \mu_A(x)) | x \in U\} \tag{1}$$

where $\mu_A: x \rightarrow [0,1]$ is the membership function, $\mu_A(x)$ is the degree of membership of the variable x, and U is the domain of the application, called in "fuzzy" terms the universe in discourse. The closer $A$ is to the value of 1, the greater the relevance of the object x to the set $A$.

- Membership functions: These represent the degree to which an element belongs to a subset defined by a label. They allow us to represent a fuzzy set graphically. The x-axis

(abscissa) represents the universe in discourse, while the y-axis (ordinate) represents the degrees of membership in the interval [0, 1]. To define the fuzzy set, the triangular and trapezoidal functions will be used, as shown in Table 4, because they allow for modeling uncertainty and ambiguity and for representing graphically how the degree of belonging of an element to a fuzzy set varies, which is crucial in systems where the binary classification (belonging or not belonging) is too restrictive [44].

**Table 4.** Membership functions used.

| Triangular Function | Trapezoidal Function |
| --- | --- |
| 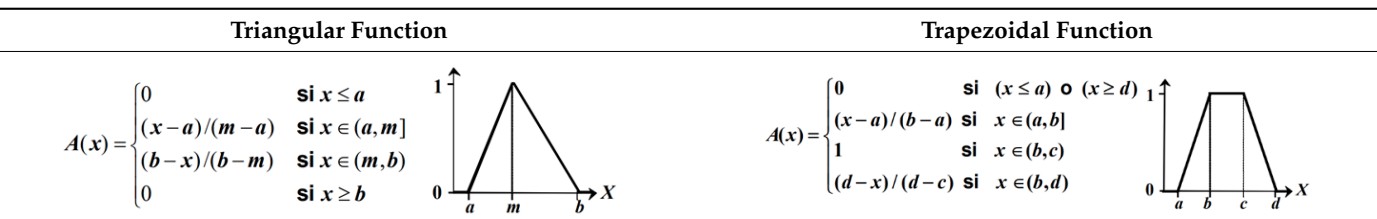 | |

- Fuzzy systems: The starting point for building a fuzzy system is to obtain a collection of fuzzy rules based on human knowledge. The next step is to combine these rules into a simple design. Different fuzzy systems employ different principles of this combination; for this purpose, the Takagi–Sugeno–Kang (TSK) fuzzy system will be used for its accuracy in modeling complex and nonlinear relationships, providing clear and accurate numerical outputs [45].
- Defuzzification: An essential mathematical process in fuzzy logic that transforms fuzzy sets into a representative actual number. Although the fuzzy inference system produces conclusions in fuzzy terms, it is required to convert this fuzzy information into a singular real value to obtain a valuable and concrete output. Among the available defuzzification methods, the centroid method will be used, which allows the calculation of the x-coordinate of the center of gravity of the resulting fuzzy set, thus providing a fundamental value that effectively represents the fuzzy conclusion [46].

## 4. Experimentation

### 4.1. Participants

This study focused on the quantitative evaluation of the usability of a mobile interface for electricity consumption management, using a method characteristic of quantitative approaches [47,48], which opts for specific participants that best address the research questions without imposing gender restrictions, thus ensuring adequate representativeness within the national demographics. For this case, we focused on residents of the Republic of Panama of legal age with access to electric power, electronic devices, and the ability to interact with mobile applications.

The strategy ensures the capture of representative data, which is fundamental for the statistical analysis and applicability of the findings, reflecting the needs and responses of the collective towards technological solutions in efficient energy consumption. The approach includes a comprehensive analysis of existing applications in the field of energy management, complemented by a socio-demographic survey applied to 82 users to collect essential data on experience and usability with mobile technologies. This preliminary phase allows for establishing a frame of reference to discern the intrinsic characteristics of the sample, facilitating relevant pre- and post-intervention comparisons (see Table 5).

**Table 5.** Attributes of sociodemographic characteristics.

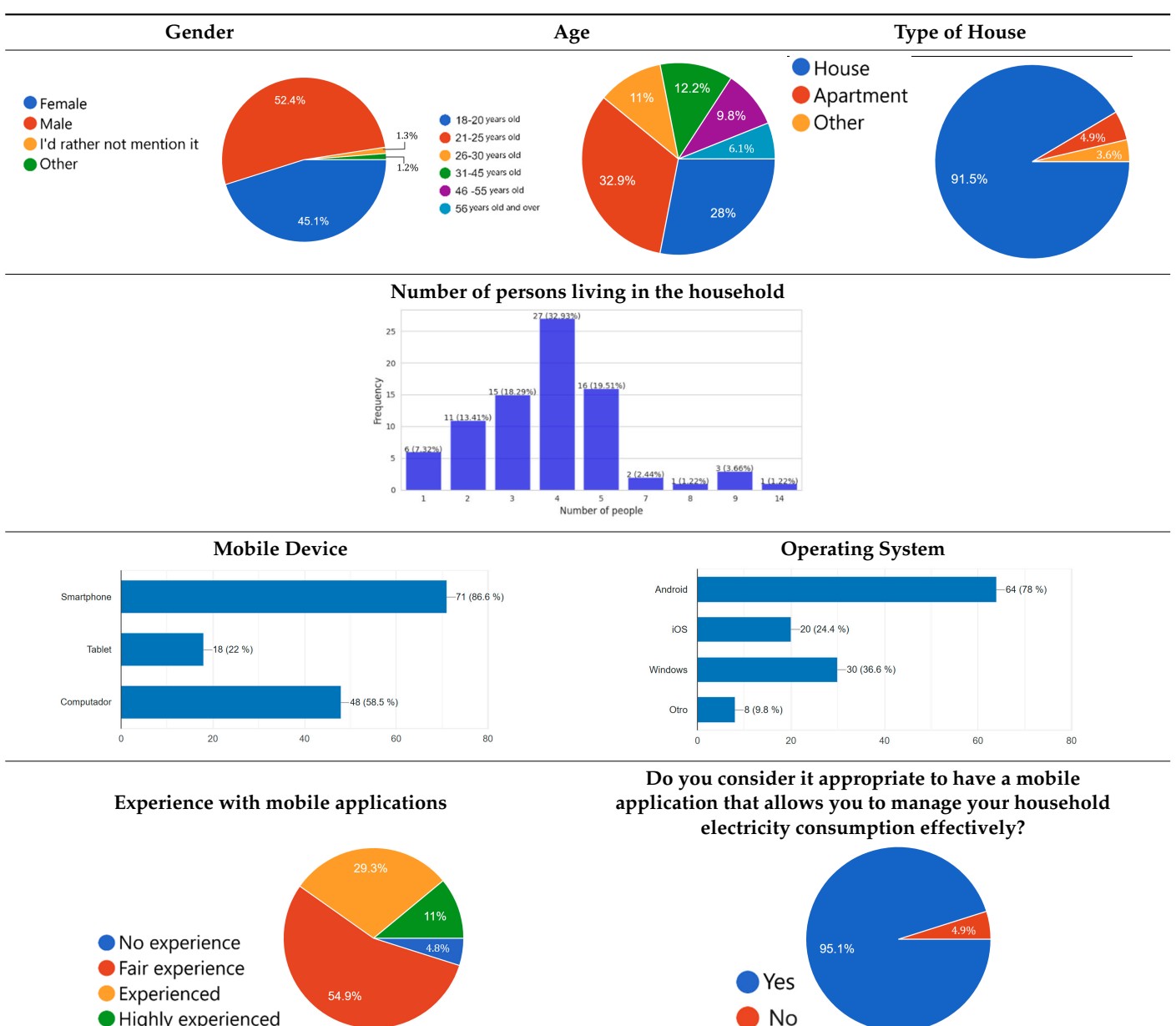

### 4.2. Requirements

With the results of the previous phase, the essential features of the mobile interface and the technological tools used by the participants were identified, and the feasibility and technical complexity of each requirement for prioritization were assessed. To assess feasibility, a literature search [23,49] on accessibility features and services was conducted. Technical complexity was measured by considering available features, services, development tools, and programming difficulty. With the MoSCoW method [50], requirements were classified on a scale of 1 to 4, where 1 indicates critical elements, and 4 indicates features not included in the current version (see Table 6). This process allowed us to objectively categorize the requirements as essential, necessary, or optional, facilitating the definition of final technical requirements for developing a functional, accessible, and technically feasible interface [51].

**Table 6.** Requirements or functional categories for the mobile interface.

| Characteristic | Description | Complexity |
|---|---|---|
| Dashboard | The main screen provides a real-time overview of energy consumption, device status, and essential alerts or messages. | 1, high |
| History and Analysis | Allows users to review their past and present energy consumption through detailed statistics and graphs. | 1, high |
| Control and Automation | Facilitates remote control of connected devices and energy systems at home. Allows certain behaviors or responses to be automated based on preferences or environmental conditions. | 2, medium |
| Settings and Preferences | Section where users can customize their profiles, alert settings, energy-saving preferences, and other general aspects. | 3, low |
| Tips and Recommendations | Provides energy-saving tips based on analysis of user behavior and energy system performance. | 3, low |
| Functions | It encompasses several additional functionalities that the interface could offer, such as consumption calculators and maintenance notifications. | 3, low |
| Help and Support | Offers user assistance through FAQs, guides, and possibly live support or a ticketing system for inquiries and troubleshooting. | 3, low |

### *4.3. Design*

#### 4.3.1. Information Architecture

To efficiently structure the content and functions of the mobile interface, we used the "card sorting" technique, a critical method in UX research, where participants organize and categorize thematic cards in a way that makes sense to them. This technique facilitates understanding user perceptions and aids in designing intuitive navigation and labeling systems. A variant, "hybrid card sorting", was applied, which combines open and closed methods, allowing both predefined categories and new ideas from participants [52,53]. With 45 participants, we ensured sufficient quantitative data for an accurate analysis, avoiding significant structural changes later in the design. The similarity matrix was thus obtained to visualize associations and consensus among participants, providing a solid basis for formulating navigation systems and menus aligned with user logic and expectations (see Figure 2).

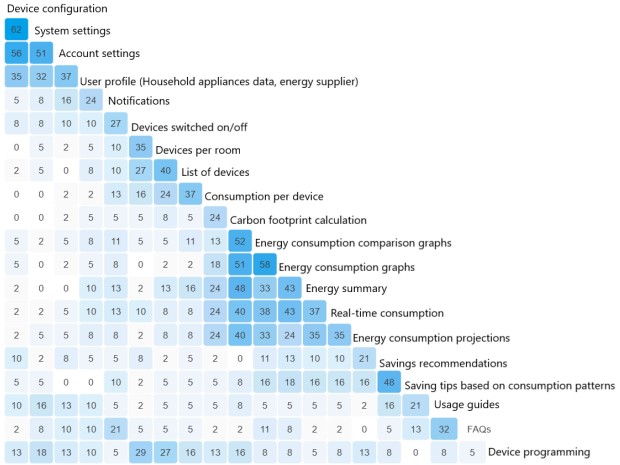

**Figure 2.** Similarity matrix.

#### 4.3.2. High-Fidelity Prototyping

High-fidelity prototypes emerge as crucial tools in the advanced stages of the design process, characterized by their close similarity to the final product in terms of design,

interactivity, and functionalities. The main advantage lies in their ability to provide a realistic and detailed representation of the end-user experience, allowing a thorough evaluation of crucial aspects such as the usability, aesthetics, and functionality of the design, by faithfully simulating the interaction and operating environment of the final product, facilitating the identification and resolution of potential problems before final implementation, thus improving the quality and effectiveness of the product [49].

During the prototyping phase for the application interface, a comprehensive and multidimensional design was adopted, covering a wide range of elements, ranging from the selection of the color palette to the creation of buttons and iconography, as shown in Figure 3. A vital aspect of this process was the incorporation of data obtained through the information architecture, which were structured and distributed by the users themselves, thus ensuring that design decisions were informed and guided by their needs and preferences, ensuring that the final design was not only aesthetically pleasing and functionally efficient but also intuitively understandable and accessible to the target audience [54].

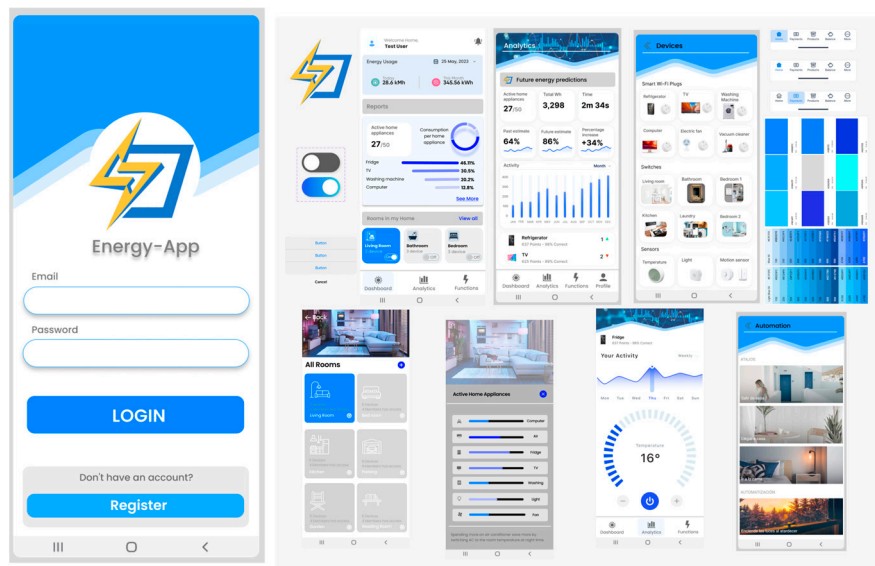

**Figure 3.** Mobile interface design.

*4.4. Evaluations*

In usability research, user-based evaluations are fundamental, and for this study, an unmoderated asynchronous remote usability test was applied to 75 users [55]. This method allows users to interact with the interface in their environment and time. It uses third-party software to facilitate efficient data collection from a wider group of users and provide realistic and objective feedback [56,57]. During the test, metrics were previously defined (see Table 7), and five specific tasks were established for users to complete questionnaires to measure their satisfaction with the interface's user experience. These processes provide a detailed understanding of the user's ease of use, effectiveness, efficiency, and satisfaction with the evaluated system [58].

**Table 7.** Metrics used in this study.

| Metrics |
|---|
| • Success and Failure: Percentage of success or failure per task of the users. Literally, how many users could perform the job correctly or if the user could not perform the task (effectiveness). |
| • Time: Each attempt to solve a task or time per task before success or failure (efficiency). |
| • Behavior: Number of errors made by users, most repeated errors, number of clicks, number of keystrokes, number of touches. |
| • Satisfaction with the interface and user experience. |

### 4.4.1. Success and Failure Metrics

Concerning the percentage of successes and failures in the study, 70% of the users completed the test, while 30% did not or abandoned the test, as shown in Figure 4.

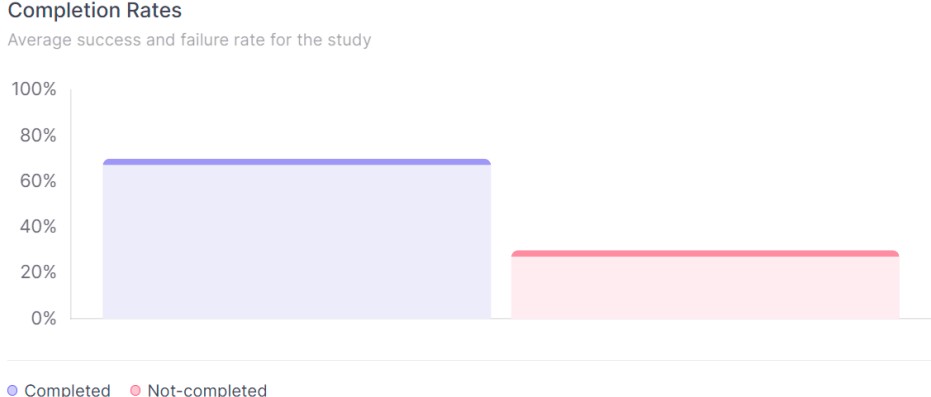

**Figure 4.** Completion rates (average success and failure rate).

### 4.4.2. Time Metrics

In Figure 5, we observe the total time users spent on the study, with the frequency of users divided into time intervals. Most users who completed the study spent between 0 s and 5 min 36 s on it; this might suggest that those who spend time on the survey are more likely to complete it, indicating that the study is designed in such a way that it can be completed in a shorter period. The presence of users who did not complete the survey at each time interval suggests that there are points of difficulty or disinterest that led users to drop out of the study.

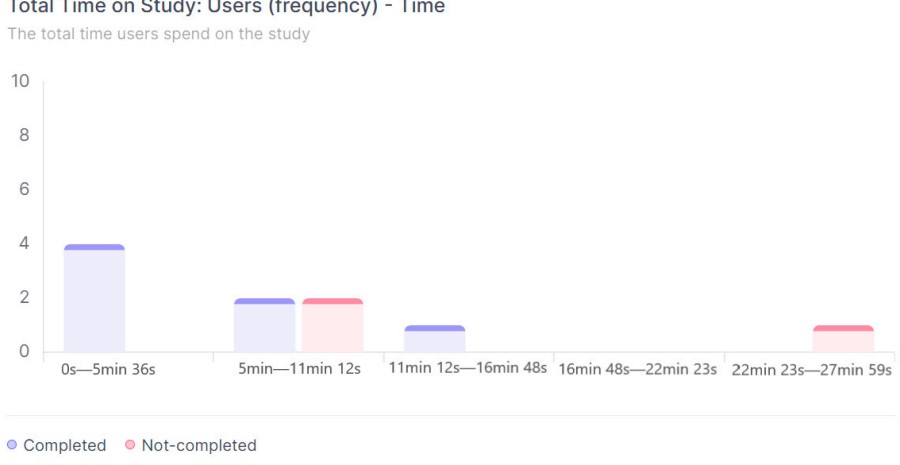

**Figure 5.** Total time users spent on the study.

In addition, we must convert the raw data into statistical metrics to make sense of the data and to detect patterns. Thus, the mean time to complete the task, the median time to complete the task, the number of users above or below, and the standard deviation of the time to complete the task with the percentage of users who completed the task correctly were calculated. Table 8 shows the statistical metrics obtained for time and errors during the study.

Times

- Average Times: The average time is approximately 797.51 s. This suggests that, on average, users take about 13 min and 29 s to complete the task or activity in question.

- Median Time: The median time is approximately 759.28 s. This indicates that half of the users take less than 12 min and 65 s to complete the task, while the other half take longer.
- Mode of Times: The mode of the times is 0:10.62 s. This could indicate a group of users who do not complete the task.
- Standard Deviation of Times: The standard deviation of the times is approximately 531.25 s. This suggests that there is significant variability in user times. Some users may be very fast, while others may be slower.

Errors

- Average Number of Errors: The average number of errors is approximately 2.19. This indicates that, on average, each user makes about two errors in completing the task.
- Median of Errors: The median of the errors is approximately 2. This suggests that half of the users make two or fewer errors while the other half make more.
- Standard Deviation of Errors: The standard deviation of the number of errors is approximately 1.77. This suggests that the number of errors made by users can vary significantly. Some users may make very few errors, while others may make more errors.

**Table 8.** Descriptive statistics of times and errors.

| Statistics | Time (s) | Errors |
|---|---|---|
| Mean | 797.51 s | 2.19 |
| Median | 759.28 s | 2.00 |
| Mode | 0:10.62 s | 0 |
| Standard deviation | 531.25 s | 1.77 |
| Task times | Minimum: 10.62 s<br>Maximum: 29 min, 54.5 s | 0 |
| Erroneous click rate | 0 | 65% |

These statistics suggest a wide variability in the timing and number of user errors. Some users are faster and more accurate, while others may take longer or make more errors. Figure 6 shows a histogram with a fitting curve representing the frequency distribution of times, measured in seconds, visualizing the distribution of time participants have taken in some activity, with key indicators of central tendency.

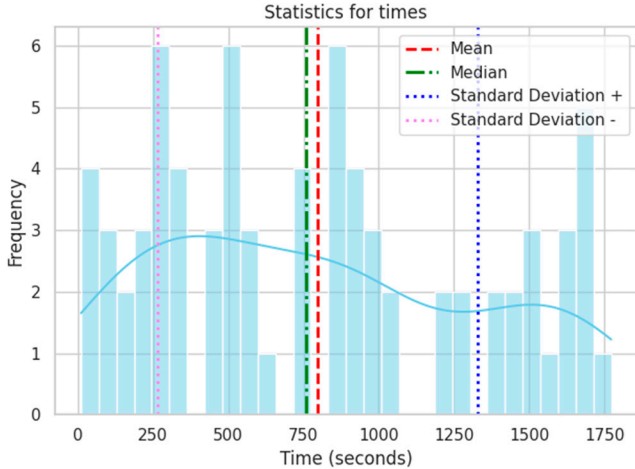

**Figure 6.** Total time users spend studying.

### 4.4.3. Behavioral Metrics

In analyzing user behavior, the heat mapping technique has been used, which allows the visualization of data showing the intensity of the information using colors [59,60]. Figure 7 shows user interactions by indicating where they touch most frequently on the screen; areas with warmer colors (reds, yellows) indicate more interaction, while cooler-colored areas (blues, greens) indicate less interaction. This makes it possible to understand which application areas are most used and which might need improvement, either to increase interactivity or to make them more accessible and optimize the application's design, ensuring that essential functions are easy to find and use. This analysis helps improve the user experience and make data-driven decisions on improving the application.

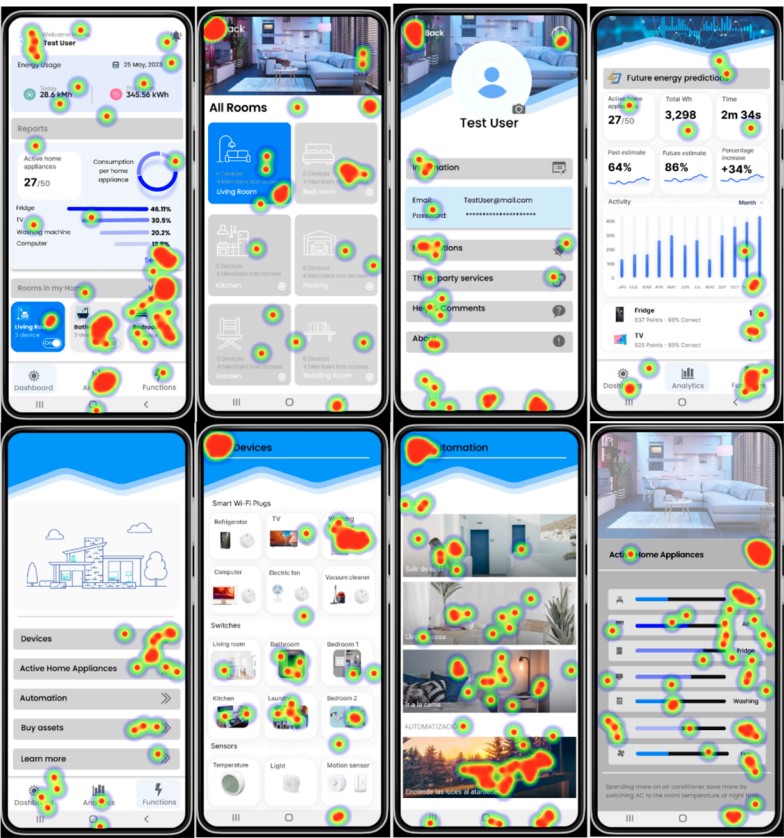

**Figure 7.** Analysis of user interaction on the interface.

### 4.4.4. Interface Satisfaction and User Experience Metrics

The choice of UEQ-S and SUS as metrics for evaluating user experience and usability is justified by their ability to provide fast and effective evaluations of interactive products. The UEQ-S, with its focus on pragmatic and hedonic qualities, provides a holistic view of user experience, encompassing both usability and user experience goals. On the other hand, the SUS is a widely recognized and established tool that allows usability issues to be identified efficiently through its scoring scale and its focus on specific questions reflecting usability and feature integration [55,61,62]. Both tools are complementary and provide a holistic view of the user's interaction with the product, making them suitable for the objectives of this study that seek to assess both usability and user experience. In addition, the combination of quantitative and qualitative analysis offered by these metrics allows for a deeper understanding of user opinions and potential usability issues [33,63].

The SUS and UEQ-S were used to assess user satisfaction with the interface and verify whether it meets usability standards. Users were organized into two groups, with 30 completing the SUS and 45 answering the UEQ-S. The SUS provides a comprehensive

score reflecting the overall usability of the system. The results, shown in Figure 8, break down the total satisfaction percentage obtained on each SUS questionnaire question.

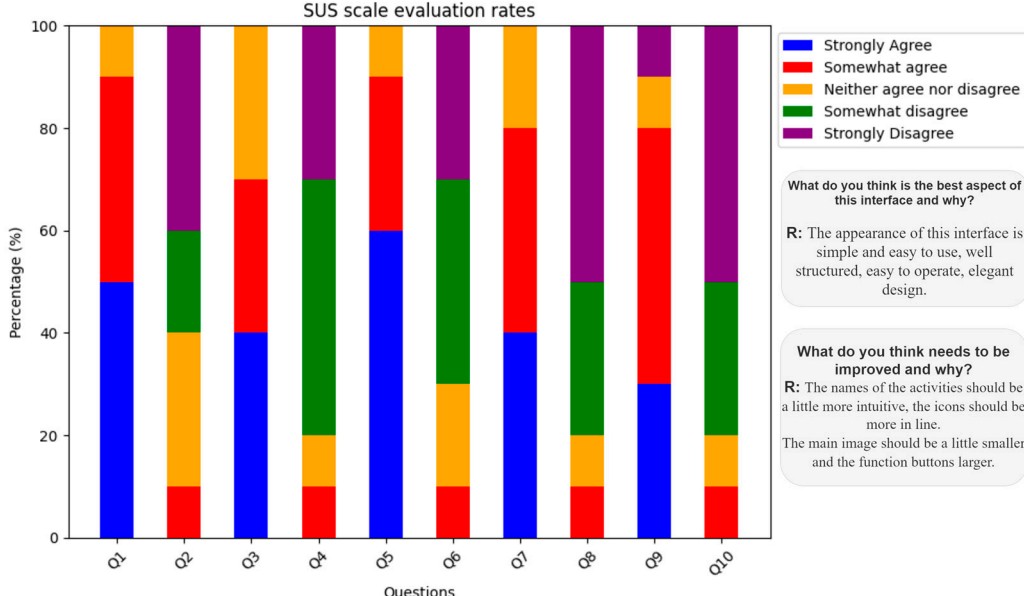

**Figure 8.** SUS percentage results.

For the UEQ-S, the means and consistency of the pragmatic quality and hedonic quality scales are interpreted. The UEQ-S results indicate that both pragmatic and hedonic quality are positive, with an average value above 1 (see Table 9).

**Table 9.** Average scores obtained.

| Short UEQ Scales | | |
|---|---|---|
| Pragmatic Quality | ⬆ | 1.244 |
| Hedonic Quality | ⬆ | 1.144 |
| Overall | ⬆ | 1.194 |

A value of 1.244 suggests that users find the interface to have excellent pragmatic quality, i.e., it fulfills its purpose effectively and efficiently. At the same time, a score of 1.144 indicates positive hedonic aspects, i.e., it evaluates the user experience, including factors such as satisfaction, enjoyment, and visual stimulation. This suggests that users are generally satisfied with the usability and aesthetic pleasure of the user experience (see Figure 9).

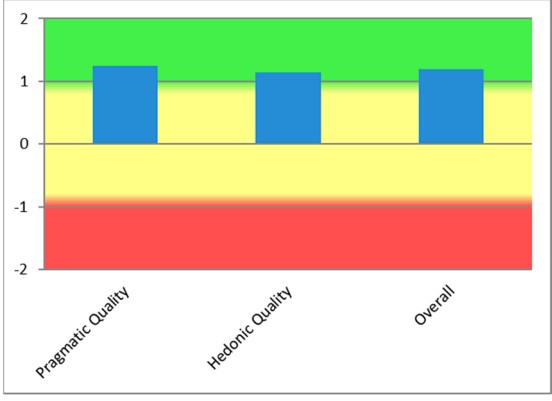

**Figure 9.** Scores for both scales.

The corresponding values of Cronbach's Alpha [64] were 0.84 (pragmatic quality) and 0.81 (hedonic quality), as seen in Table 10, suggesting that both scales have high reliability and that the items of each scale are consistently related to each other.

**Table 10.** Item correlations by scale and Cronbachs alpha coefficient.

| Pragmatic Quality | | Hedonic Quality | |
|---|---|---|---|
| **Items** | **Correlation** | **Items** | **Correlation** |
| 1.2 | 0.35 | 5.6 | 0.69 |
| 1.3 | 0.68 | 5.7 | 0.23 |
| 1.4 | 0.52 | 5.8 | 0.53 |
| 2.3 | 0.53 | 6.7 | 0.47 |
| 2.4 | 0.79 | 6.8 | 0.59 |
| 3.4 | 0.53 | 7.8 | 0.54 |
| **Average** | 0.57 | **Average** | 0.51 |
| **Alpha** | **0.84** | **Alpha** | **0.81** |

## 5. Fuzzy Model

To analyze and validate the SUS and UEQ-S results in a way that adequately captures the variability and trends in user responses, the fuzzy logic model was used in this study. To implement this model, we used Python [65,66] with the scikit-fuzzy library, in which we created antecedent and consequent objects for the variables, defined the membership functions with methods such as trim (triangular function) or trapmf (trapezoidal function), configured the control system with the inference rules, and, finally, executed the model to obtain de-fuzzy results that provide the probability of recommendation and the quality of the user's experience in clear values. The following are the steps.

1. Definition of Variables and Fuzzy Sets

The variables that would be part of the fuzzy model, in the context of SUS and UEQ-S, were defined. Each of these variables was modeled as a fuzzy set with degrees of membership ranging from 0 to 1. We have the following:

Input Variables (SUS- and UEQ-S-based).

- Ease of use (SUS): $F(x)$ [0, 101, 1]
- Satisfaction (UEQ-S): $S(x)$ [−3, 4, 1]
- Efficiency (average task time): $E(x)$ [0, 601, 1]

Each variable is associated with a membership function, which indicates the degree to which a specific value belongs to the fuzzy set. These functions are represented as

$$\mu_F(x) : x \rightarrow [0, 1]$$

$$\mu_S(x) : x \rightarrow [0, 1]$$

$$\mu_E(x) : x \rightarrow [0, 1]$$

where $\mu_F(x)$, $\mu_S(x)$, $\mu_E(x)$ are the membership functions for Ease of Use, Satisfaction and Efficiency, respectively, and x is the specific value of each variable.

Output Variables (Objective of the Analysis)

- Probability of recommendation: $P(y)$ [0, 101, 1]
- Overall quality of user experience: $Q(y)$ [0, 101, 1]

Here, $\mu_P(y)$, $\mu_Q(y)$ represent the membership functions for the probability of recommendation and overall quality of user experience, respectively, and y is the specific value of each output variable.

For the input variable "ease of use", values range from 0 to 100, with increments of 1. This means that ease of use can be rated on a scale of 0 to 100, where 0 represents a very low ease of use and 100 represents a very high ease of use.

For the input variable "satisfaction", values range from $-3$ to 3, with increments of 1. This indicates that satisfaction can be rated on a scale of $-3$ to 3, where negative values represent dissatisfaction, 0 represents neutrality, and positive values represent satisfaction.

For the input variable "efficiency", values range from 0 to 600, with increments of 1. This suggests that efficiency can be measured on a scale of 0 to 600, where higher values indicate greater efficiency.

For the output variables "probability of recommendation" and "UX quality", the values also range from 0 to 100, with increments of 1. This means that both the likelihood of recommendation and the quality of the user experience can be evaluated on a scale from 0 to 100, where higher values indicate a higher likelihood of recommendation or a higher quality of the user experience.

2. Creating Membership Functions

For each input variable, we define or create membership functions that describe how the degrees of membership are assigned to the values of the variable, and we use mathematical functions that represent the descriptions, as shown in Figure 10 and Table 11.

**Table 11.** Description of variable labels.

| Variables | Label | Description of the Membership Function |
|---|---|---|
| Ease of Use | Low | It would cover SUS scores from 0 to around 60, with full membership at 0 and decreasing linearly until reaching 60, where membership would be 0. $$\mu_{Fbaja}(x) : \max\left(0, min\left(1, \frac{60-x}{60}\right)\right) para\ 0 \leq x \leq 60$$ |
| | Medium | It would be centered around the average score of 50, starting to increase from 40, peaking at 60, and then decreasing to 80. $$\mu_{Fmedia}(x) : \max\left(0, min\left(\frac{x-40}{20}, 1, \frac{80-x}{20}\right)\right) para\ 40 \leq x \leq 80$$ |
| | High | It increases its degree of membership from around 60 and would reach full membership at 80 and remain so until 100. $$\mu_{Falta}(x) : \max\left(0, min\left(1, \frac{x-60}{40}\right)\right) para\ 60 \leq x \leq 100$$ |
| Satisfaction | Dissatisfied | Full membership would be at $-3$ (very dissatisfied) and decrease to 0. $$\mu_{Sinsatisfecho}(x) : \max\left(0, min\left(1, \frac{-x+3}{3}\right)\right) para -3 \leq x \leq 0$$ |
| | Neutral | This function would gradually increase from $-1$ until reaching full membership at 0 and decreasing again towards 1. $$\mu_{Sneutral}(x) : \max\left(0, min\left(\frac{x+1}{1}, 1, \frac{1-x}{1}\right)\right) para -1 \leq x \leq 1$$ |
| | Satisfied | It would start to increase at 0 and reach full membership at 2 and would be maintained until 3. $$\mu_{Ssatisfecho}(x) : \max\left(0, min\left(1, \frac{x}{2}\right)\right) para\ 0 \leq x \leq 3$$ |
| Efficiency | Low | A long time to complete a task (say more than 200 s) would have a high membership at low efficiency. $$\mu_{Ebaja}(x) : \max\left(0, min\left(1, \frac{500-x}{300}\right)\right) para\ 200 \leq x$$ |
| | Medium | A moderate task time (between 100 and 500 s) would be associated with medium efficiency. $$\mu_{Emedia}(x) : \max\left(0, min\left(\frac{x-100}{400}, 1, \frac{500-x}{400}\right)\right) para\ 100 \leq x \leq 500$$ |
| | High | A short task time (less than 300 s) would indicate high efficiency. $$\mu_{Ealta}(x) : \max\left(0, min\left(1, \frac{300-x}{300}\right)\right) para\ x \leq 300$$ |
| Probability of Recommendation | Low | If ease of use and/or satisfaction are low, the likelihood of recommendation would be low. |
| | Medium | If both ease of use and satisfaction are moderate, the likelihood of recommendation would be medium. |
| | High | If both ease of use and satisfaction are high, the likelihood of recommendation would be high. |

**Table 11.** *Cont.*

| Variables | Label | Description of the Membership Function |
|---|---|---|
| | Low | Low ease of use, satisfaction, and efficiency scores would lead to low overall quality. |
| Overall Quality of User Experience | Medium | Medium scores in these areas would result in a medium overall quality of experience. |
| | High | High scores in ease of use, satisfaction, and efficiency would indicate a high overall quality of user experience. |

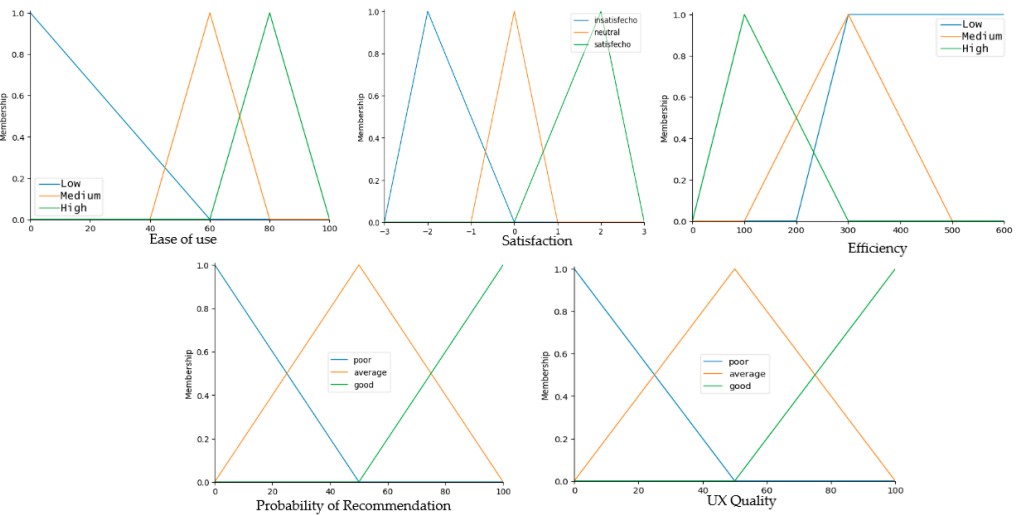

**Figure 10.** Plots of membership functions.

3.  Fuzzy Inference Rules

Based on the membership functions, we define rules to infer the probability of recommendation and the overall quality of user experience:

- If ease of use is high and satisfaction is high, then the likelihood of recommendation is high.

$$Rule\ 1 = \text{if } \min\left(\mu_{F_{high}}(x),\ \mu_{S_{high}}(x)\right) \text{ then } \mu_{P_{high}}(y)$$

where $\mu_{F_{high}}(x)$ and $\mu_{S_{high}}(x)$ are the membership functions for ease of use and satisfaction, respectively, evaluated at the "high" level.

- If ease of use is low or satisfaction is low, then the overall quality of experience is low.

$$Rule\ 2 = \text{if } \max\left(\mu_{F_{low}}(x),\ \mu_{S_{low}}(x)\right) \text{ then } \mu_{Q_{low}}(y)$$

where $\mu_{F_{low}}(x)$ and $\mu_{S_{low}}(x)$ are the membership functions for ease of use and satisfaction, respectively, evaluated at the "low" level.

- If efficiency is low, then the overall quality of experience is low, regardless of the ease of use and satisfaction.

$$Rule\ 3 = if\ \mu_{E_{low}}(x) \text{ then } \mu_{Q_{low}}(y)$$

where $\mu_{E_{low}}(x)$ is the membership function for efficiency evaluated at the "low" level.

- If the ease of use is high, satisfaction is high, and efficiency is high, then the quality of the user experience is good.

$$Rule\ 4 = if\ \mu_{F_{high}}(x)\ And\ \mu_{S_{high}}(x)\ And\ \mu_{E_{high}}(x) \text{ then } \mu_{Q_{good}}(y)$$

where $\mu_{F_{high}}(x)$ and $\mu_{S_{high}}(x)$ are the membership functions for ease of use and satisfaction, respectively, evaluated at the "high" level.

- If the ease of use is low or satisfaction is low, then the quality of the user experience is poor.

$$Rule\ 5 = if\ \mu_{F_{high}}(x)\ Or\ \mu_{S_{low}}(x)\ then\ \ \mu_{Q_{poor}}(y)$$

where $\mu_{F_{high}}(x)$ and $\mu_{S_{low}}(x)$ are the membership functions for ease of use and satisfaction, respectively, evaluated at the "low" level.

4. Implementing Fuzzy Inference

A fuzzy inference engine is used to apply the fuzzy rules to the input data and practices to arrive at a fuzzy conclusion. This involves combining the membership functions of the input variables using fuzzy operators (AND, OR, NOT) and then applying an inference method, in this case the Takagi–Sugeno.

After defining the input variables, output variables, and rules, we create a control system using skfuzzy's ControlSystem class. This control system is essentially a collection of rules that define how inputs should be combined to produce outputs.

**# Creating the control system**
**control_system = ctrl.ControlSystem([rule1, rule2, rule3, rule4, rule5])**

Next, we create a ControlSystemSimulation object using the control system we defined. This object allows us to input specific values for the antecedents (input variables) and compute the consequents (output variables) based on the defined rules.

**control_simulation = ctrl.ControlSystemSimulation(control_system)**

Now, we can set the input values for the antecedents using the input property of the ControlSystemSimulation object. These input values represent the specific conditions or factors we want to evaluate using the fuzzy inference engine.

**# Set input values**
**control_simulation.input['Ease of Use'] = 80**
**control_simulation.input['Satisfaction'] = 2**
**control_simulation.input['Eficciency'] = 250**

To compute the output values (consequents), we use the compute() method of the ControlSystemSimulation object. This method applies the fuzzy rules to the input values and calculates the corresponding output values.

**# Compute output values**
**control_simulation.compute()**

After computing the output values, we can access them using the output property of the ControlSystemSimulation object. These output values represent the fuzzy conclusions drawn from the input data and rules.

**# Get output values**
**output_probabilidad_recomendacion = control_simulation.output['Probability of Recomendation]**
**output_calidad_ux = control_simulation.output['UX Quality']**

5. Defuzzification

Finally, the fuzzy output is converted into a crisp or precise value that is understandable and useful. This is performed by defuzzification processes such as the centroid method, which gives a single value representing the fuzzy inference result, as shown in Equation (2) and with which the values shown in Figure 11 are obtained.

$$y_d = \frac{\int_s y\mu_Y(y)dy}{\int_s \mu_Y(y)dy} \tag{2}$$

where $\mu_Y$ is the membership function of the output set $Y$, whose output variable is $y$, and $s$ is the domain or range of integration.

```
Universe of Probabilidad de Recomendacion Consequent: [  0   1   2   3   4   5   6   7   8   9  10  11  12  13  14  15  16  17
 18  19  20  21  22  23  24  25  26  27  28  29  30  31  32  33  34  35
 36  37  38  39  40  41  42  43  44  45  46  47  48  49  50  51  52  53
 54  55  56  57  58  59  60  61  62  63  64  65  66  67  68  69  70  71
 72  73  74  75  76  77  78  79  80  81  82  83  84  85  86  87  88  89
 90  91  92  93  94  95  96  97  98  99 100]
Output of Control Simulation for Probabilidad de Recomendacion Consequent: 83.33333333333336
Universe of Calidad UX Consequent: [  0   1   2   3   4   5   6   7   8   9  10  11  12  13  14  15  16  17
 18  19  20  21  22  23  24  25  26  27  28  29  30  31  32  33  34  35
 36  37  38  39  40  41  42  43  44  45  46  47  48  49  50  51  52  53
 54  55  56  57  58  59  60  61  62  63  64  65  66  67  68  69  70  71
 72  73  74  75  76  77  78  79  80  81  82  83  84  85  86  87  88  89
 90  91  92  93  94  95  96  97  98  99 100]
Output of Control Simulation for Calidad UX Consequent: 77.97619047619047
The universes do not match.
```

**Figure 11.** Defuzzification by the centroid method.

Steps to calculate the centroid:

- Weighting: each value of the output variable is multiplied by the corresponding membership function.
- Sum of weighted products: all the products obtained in the previous step are added together.
- Sum of the membership functions: all the membership functions are added together.
- Division: the sum of weighted products is divided by the sum of the membership functions.

To validate the fuzzy logic model, sensitivity tests were performed by adjusting the model parameters and observing how they affected the results. In addition, the results of the model were compared with benchmark metrics to demonstrate its accuracy and reliability in predicting the probability of recommendation and quality of user experience.

## 6. Results

To analyze the results of the usability and user experience tests, we first organized the SUS scores into a percentile chart. We summed the mean scores from the user questionnaire, and with a mean score of 68, we reached a total score of 80 points, as shown in Figure 12. This result reflects a very satisfactory user experience with the interface, indicating an acceptable and promising margin.

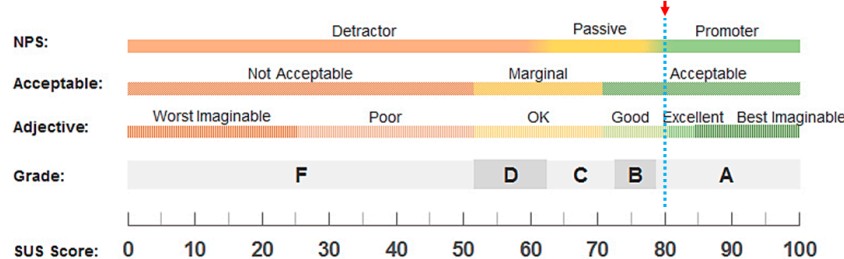

**Figure 12.** Percentage representation of SUS results.

On the other hand, the UEQ-S results reveal a generally positive experience with the evaluated energetic mobile interface, exceeding the average. The scores indicate a high degree of satisfaction with both the usability (pragmatic quality) and the emotional and aesthetic aspects (hedonic quality) of the user experience, as seen in Figure 13.

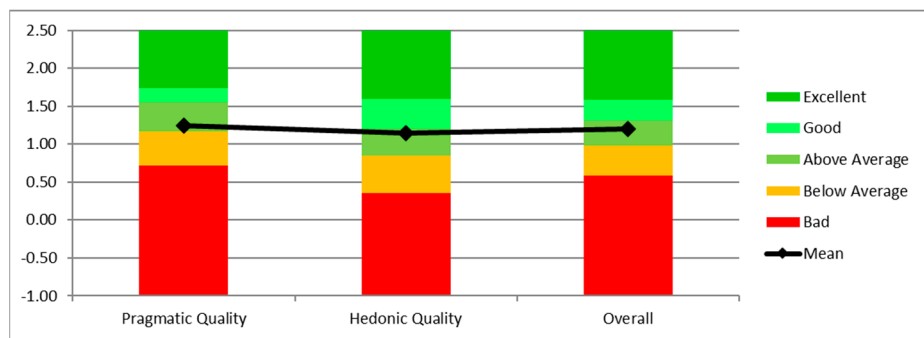

**Figure 13.** User experience quality evaluation: UEQ-S benchmarking.

Finally, in the fuzzy logic system, we have the input values for the indicators of ease of use, satisfaction, and efficiency, respectively. In the defuzzification process, the degrees of relevance of each of the input variables are calculated for each of their labels. When defuzzification is applied to the input variables, each with numerical values associated with each one of them, the results of Table 12 are obtained. The calculation of the degrees of membership is performed according to the typical membership functions.

**Table 12.** Degrees of membership of input values to the fuzzy sets.

| Entry Value | Ease of Use (Low) | Ease of Use (Medium) | Ease of Use (High) | Satisfaction (Dissatisfied) | Satisfaction (Neutral) | Satisfaction (Satisfied) | Efficiency (Low) | Efficiency (Medium) | Efficiency (High) |
|---|---|---|---|---|---|---|---|---|---|
| 80 | 0 | 0.5 | 0.5 | 0 | 0 | 1 | 0 | 0.75 | 0.25 |
| 2 | 0 | 0.5 | 0 | 0 | 0.5 | 0 | 0 | 0.5 | 0 |
| 250 | 0 | 0.5 | 0 | 0 | 0 | 0 | 0 | 0.75 | 0.25 |

The fuzzy inference process is performed through the defined rules, and once this process has been carried out, the values shown in Table 13 are obtained for the output variables.

**Table 13.** Degrees of membership of the output variable to the fuzzy sets.

| Fuzzy Set | Degree of Membership (Probability of Recommendation) | Degree of Membership (Quality of UX) |
|---|---|---|
| Low | 0.2 | 0.6 |
| Medium | 0.8 | 0.3 |
| High | 0.5 | 0.9 |

Figure 14 shows the graphs with the satisfaction results, in which the model indicates that the level of satisfaction with the interface and the ease of use are relatively high. The efficiency is shown as medium and high, which are fundamental since they define how the quantitative values are converted into fuzzy qualitative ratings. The inference rules then use these ratings to determine the system's outputs based on the given inputs. They are the visual representation of how the input maps to a degree of membership in each of the fuzzy sets defined for each variable.

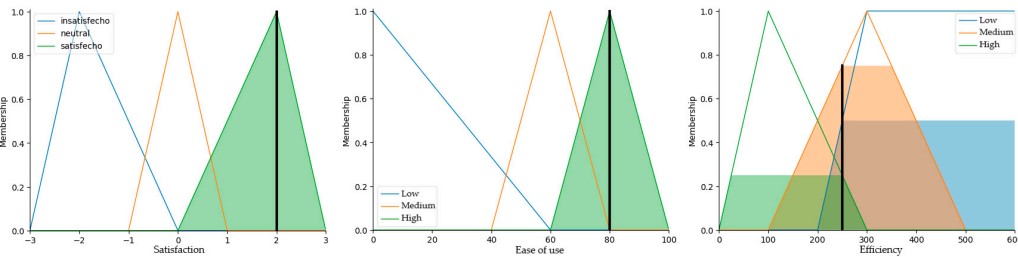

**Figure 14.** Results for input variables.

This model allows a deeper understanding of the evaluations made through the User Experience Questionnaire Short (UEQ-S) and the System Usability Scale (SUS). These numerical values are the result of the defuzzification of the fuzzy logic model and represent the composite evaluation of the system we are analyzing and how results are obtained.

- Figure 15 shows the probability of recommendation (83.33): This value suggests that there is a high probability that users will recommend the mobile interface system for residential energy management. In fuzzy logic, this is deduced from rules that associate inputs of high ease of use and high satisfaction with a high probability of recommendation. Since both ease of use and satisfaction were rated high, the fuzzy inference results in a positive rating. It could be interpreted as an indicator that users are satisfied with the system and willing to share their positive experiences with others.

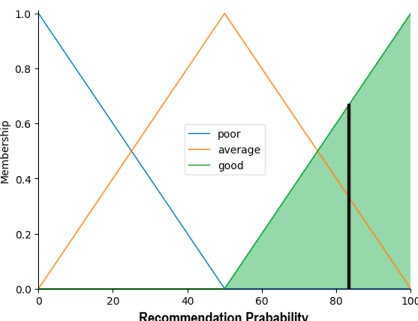

**Figure 15.** Results of the recommendation probability.

- Figure 16 shows the overall quality of user experience (77.98): This value indicates that the overall quality of user experience with the system is also high. It has been calculated considering ease of use satisfaction and the efficiency of user tasks. It suggests that users find the system easy to use and satisfying and that it allows for efficient interaction.

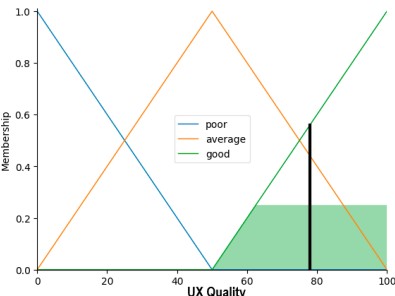

**Figure 16.** UX quality results.

This study focused on evaluating the satisfaction, ease of use, and efficiency of the mobile residential energy management interface. In terms of satisfaction, we found that users expressed high satisfaction with the overall experience, highlighting the visual appearance and novelty of the interface as emotionally satisfying aspects. In addition, the interface was reported to be efficient and easy to use in daily tasks, supporting the effectiveness of the design in terms of user satisfaction. In relation to ease of use, we observed that users praised the simplicity and clarity of the interface functions, which contributed to a positive experience in terms of usability. Finally, in terms of efficiency, users reported acceptable task times and smooth interaction with the interface, indicating that the system allows for efficient interaction to complete tasks related to residential energy management. These results will be used to continuously improve the mobile interface system for residential energy management and future system development.

## 7. Discussions

During this study, important insights about the user experience with the residential mobile energy management interface were revealed through a comparative analysis between quantitative and qualitative data. SUS scores were used to identify discrepancies between user perception and behavior during use. In addition, SUS scores were compared with UEQ-S scores to analyze the alignment between subjective perceptions and objective measures of efficiency. A summary of this comparative analysis is presented in Table 14 below.

**Table 14.** Comparative analysis of quantitative and qualitative data.

| Aspect | Quantitative Data | Qualitative Data |
|---|---|---|
| Usability | The average score of the Usability Evaluation System (SUS) was 80, indicating a very satisfactory user experience. | Users expressed high satisfaction with the ease of use of the interface, highlighting the simplicity and clarity of the functions. |
| User Experience | The average User Experience Quality Scale (UEQ-S) score was 1.194, indicating a positive experience in both pragmatic and hedonic aspects. | Users reported an emotionally satisfying experience, highlighting the visual appeal and novelty of the interface. In addition, they found the interface efficient and easy to use in their daily tasks. |
| Reliability | The internal consistency of the pragmatic and hedonic quality scales, assessed by Cronbach's Alpha coefficient, was high (0.84 for pragmatic quality and 0.81 for hedonic quality), indicating reliability of the results. | Users demonstrated consistency in their responses in both pragmatic and hedonic aspects, which supports the consistency of the findings obtained through different evaluation methods. |

These findings and approaches are relevant in the broader context of research on user experience and the adoption of innovative technologies. Through a comparative analysis with previous research, such as Yan et al. [67], Martin et al. [68], Soetedjo et al. [69], and Gardas et al. [70], an emerging trend towards improving user experience through the integration of technologies such as the Internet of Things (IoT) and fuzzy logic is observed. These studies not only explore specific applications of these technologies, but also provide a deeper understanding of how the interaction between users and technology can be optimized to improve efficiency, satisfaction, and overall user experience.

On the other hand, by adopting a user-centric approach, the results of this study offer important practical implications for the design and development of mobile residential energy management applications. In terms of design recommendations, this study suggests some key areas of focus to further optimize the mobile interface, for example, considering the SUS findings, where specific aspects of the interface were identified that could be perceived as complex or inefficient by some users, and designers could prioritize the simplification of navigation and clarity of functions to improve overall usability. In addition, the UEQ-S results highlight the importance of hedonic dimensions of user experience, such as attractiveness and novelty. Therefore, designers could consider integrating attractive visual elements and innovative features into the interface to increase user satisfaction and engagement.

Another recommendation derived from this study is the implementation of continuous feedback tools, such as application-integrated satisfaction surveys or direct feedback features, to capture user feedback and make iterative adjustments to the design over time.

Finally, this study presents encouraging results on the likelihood of user recommendation and the overall quality of user experience with the mobile interface for residential energy management. However, it is critical to consider the following: Although an effort was made to select representative participants and mixed methods were applied to gain a comprehensive understanding of the user experience, the specific demographic characteristics of the research participants may limit the extrapolation of the results to other countries or regions with different cultural and socioeconomic contexts. In addition, although quantitative and qualitative methods were employed to assess user experience,

it is important to recognize that the perception of usability and user satisfaction may be influenced by subjective and contextual factors that were not fully captured in this study. For example, cultural preferences, level of familiarity with the technology, and individual differences in aesthetic perception could influence the evaluation of the mobile interface.

## 8. Conclusions

This study explored the relationship between mobile interface design and user satisfaction in managing residential energy consumption through a mobile application. The findings indicate that there is a clear influence of interface design on user satisfaction, as evidenced by high scores on the SUS (80) and UEQ-S scales (1.244 for pragmatic quality and 1.144 for hedonic quality), reflecting a satisfying and enriching experience for users.

It is important to note that the attention devoted to user-centered design has been instrumental in ensuring the efficiency and effectiveness of the system, which is reflected in the high user ratings on pragmatic and hedonic aspects of quality of experience. In addition, elements such as ease of use, efficiency, and intuitive design are found to contribute significantly to user satisfaction and improved usability.

By applying the fuzzy logic model to analyze the results, a deeper understanding of how usability metrics relate to efficiency, effectiveness, and overall user satisfaction is obtained. The high values obtained for likelihood of recommendation (83.33) and overall quality of user experience (77.98) support the effectiveness and satisfaction of the design for users.

In relation to the research questions posed, it is shown that interface design positively influences user satisfaction and that certain design elements have a significant impact on user experience. Furthermore, a positive correlation is observed between usability metrics and user satisfaction in the use of the mobile application for residential energy management.

This study highlights the relevance of a user-centric mobile interface design to improve the residential energy management experience. However, it is important to recognize limitations, such as possible biases in participant selection, context specificity, measurement of variables, and the limited duration of the study. For future research, we suggest replicating the study in different contexts and with larger samples, conducting longitudinal studies to assess long-term effects, employing qualitative methods to better understand user experiences, conducting controlled experiments to investigate specific design elements, and exploring interdisciplinary collaborations to enrich understanding of the topic.

**Author Contributions:** Conceptualization, I.N.; methodology, I.N.; validation, I.N., E.E.C. and E.C.; formal analysis, I.N., D.C. and N.N.; investigation, I.N.; data curation, I.N. and D.C.; writing—original draft preparation, I.N.; writing—review and editing, C.R., E.E.C. and E.C.; supervision, C.R. All authors have read and agreed to the published version of the manuscript.

**Funding:** This research was funded by Secretaría Nacional de Ciencias y Tecnología Panamá, grant number: FISC-MCCM-08-2022.

**Institutional Review Board Statement:** The study was conducted in accordance with the Declaration of Bioethics and was approved by the Institutional Research Bioethics Committee, Universidad Tecnológica de Panamá (No. P-CIBio-024-2023 on 17 July 2023) for human studies.

**Informed Consent Statement:** Informed consent was obtained from all subjects involved in the study.

**Data Availability Statement:** The analyzed information can be found at the following link: https://github.com/ivonnecarmen07/Data_Usability, accessed on 18 February 2024.

**Acknowledgments:** I.N., D.C. and N.N are supported by a grant from the Program to Strengthen National Postgraduate Programs of the National Secretariat of Science, Technology, and Innovation (SENACYT) in the Master of Science in Mobile Computing program and E.E.C., E.C. and C.R. are supported by the National Research System of Panama (SNI).

**Conflicts of Interest:** The authors declare no conflicts of interest.

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
