# Peer review of "Improving Usability in Mobile Apps for Residential Energy Management: A Hybrid Approach Using Fuzzy Logic"

_applsci, doi:10.3390/app14051751_

Round 1

Reviewer 1 Report

Comments and Suggestions for Authors

The subject of the paper is very well represented. Methodology of research is very effective, and the research approach is clear and simple. Moreover, the findings are interesting and informative.

 Great Paper!

Author Response

Thank you so much for your positive comments on the article!
We are pleased that you found our findings interesting and informative. We strive to conduct a thorough analysis and present results that are meaningful and useful to the academic and professional community. We greatly value your feedback and your comments motivate us to continue to improve our research. 

Reviewer 2 Report

Comments and Suggestions for Authors

The main question addressed by the research is the application of fuzzy logic to the questionnaire data.

1. This research uses the fuzzy logic to analyze the questionnaire data. It has the originality to some extent.

2. This research needs to add the research hypotheses and the verification of these hypotheses, because it constructs the research model in the beginning.

3. We need the basic descriptive statistics about the questionnaire data. The authors can show the data in the appendix.

- I think the usage of fuzzy logic analysis has some originality to some extent; however, it needs to be elaborated. Additionally, the paper addresses a specific gap in the field related to the application of fuzzy logic in analyzing questionnaire data. 

Compared with other published journal papers, it sheds some enlightenment to the readers, adding valuable insights to the subject area.

- The authors need to emphasize the construction of research hypotheses and provide a clearer explanation of fuzzy logic analysis results to determine the validity of the research hypotheses. Additionally, further controls should be considered for a more comprehensive methodology.

- I do not find any indication of consistency or inconsistency between the conclusions and the evidence and arguments presented. Furthermore, there is no information on whether all the main questions posed were addressed and by which specific experiments.

- Basically, the references are appropriate.

The presentation of Table 8 needs to be improved.

Author Response

For research article

Response to Reviewer 2 Comments

1. Summary

Thank you very much for taking the time to review this manuscript. We extend our gratitude for your diligent review of our manuscript and for offering invaluable feedback. We appreciate your recognition of the contribution our paper makes.

2. Questions for General Evaluation

Reviewer’s Evaluation

Response and Revisions

Does the introduction provide sufficient background and include all relevant references?

Yes

[Please give your response if necessary. Or you can also give your corresponding response in the point-by-point response letter. The same as below]

Are all the cited references relevant to the research?

Yes

Is the research design appropriate?

Can be improved

Are the methods adequately described?

Can be improved

Are the results clearly presented?

Can be improved

Are the conclusions supported by the results?

Yes

3. Point-by-point response to Comments and Suggestions for Authors

Comments 1: This research uses the fuzzy logic to analyze the questionnaire data. It has the originality to some extent.

Response 1: Yes, thank you for your comments. Previous work was considered and an improved approach to this was introduced. Your feedback on the clarity of our approach and the interest of our conclusions is greatly appreciated. We're thrilled you enjoyed the article!.

Comments 2: This research needs to add the research hypotheses and the verification of these

 hypotheses, because it constructs the research model in the beginning.

Response 1: Agree. Yes, thank you for your comments. Hypothesis and research questions have been included considering your observation.

Comments 3: We need the basic descriptive statistics about the questionnaire data. The authors can show the data in the appendix.

Response 3: Thank you for your comment. A Wiki was created which contains the basic descriptive statistics on the questionnaire data. It can be accessed through the link provided in the Appendix.

Data Availability Statement: The analyzed information can be found at the following link: https://github.com/ivonnecarmen07/Data_Usability

Comments 3.1: I think the usage of fuzzy logic analysis has some originality to some extent; however, it needs to be elaborated. Additionally, the paper addresses a specific gap in the field related to the application of fuzzy logic in analysing questionnaire data.

Response of comments 3.1: Yes, thank you for your comment. Previous work was considered and an improved approach to this was introduced.

Comments 3.2: Compared with other published journal papers, it sheds some enlightenment to the readers, adding valuable insights to the subject area.

Response of comments 3.2: Yes, thank you for your comment. We are pleased to know that our work has been perceived as a useful and enriching source of information for readers, and we hope to continue improving and expanding knowledge in this area.

Comments 3.3: The authors need to emphasize the construction of research hypotheses and provide a clearer explanation of fuzzy logic analysis results to determine the validity of the research hypotheses. Additionally, further controls should be considered for a more comprehensive methodology.

Response of comments 3.3: Yes, thank you for your comment. The research hypothesis was introduced and provides a clearer explanation of the methodology and results of the fuzzy logic analysis. Adding other controls was considered but would increase the complexity of the model and its subsequent analysis.

Comments 3.4: I do not find any indication of consistency or inconsistency between the conclusions and the evidence and arguments presented. Furthermore, there is no information on whether all the

main questions posed were addressed and by which specific experiments.

Response of comments 3.4: Thank you for your comments and constructive criticism. To address this concern, we carefully reviewed and revised the article to explicitly connect our conclusions to the evidence and arguments presented throughout the manuscript. In addition, we provide a more thorough analysis of how each of the major research questions raised in the article was addressed.

Comments 3.5: Basically, the references are appropriate.

Response of comments 3.5: Yes, thank you for your comments.

Comments 3.6: The presentation of Table 8 needs to be improved.

Response of comments 3.6: Yes, thank you for your comment. Table 8 was improved.

Reviewer 3 Report

Comments and Suggestions for Authors

Clarity of Methodology and Integration:

The paper mentions a hybrid approach using both quantitative and qualitative methods but lacks clarity on how these methods are integrated within the user-centered design framework. Providing a more detailed explanation of the integration process and the rationale behind it would enhance the paper's methodological robustness.

Fuzzy Logic Model Implementation:

While the paper introduces the use of a fuzzy logic model for interpreting and contrasting data, it lacks in-depth information on the model's design and implementation specifics. A more detailed exposition on how the fuzzy logic model was employed, including parameters and decision-making processes, is crucial for transparency and reproducibility.

Metric Selection and Justification:

The paper briefly mentions the use of metrics like UEQ-S and SUS but lacks a thorough justification for their selection. Providing a rationale for choosing these specific metrics over alternatives and discussing their appropriateness for the study's objectives would strengthen the paper's methodological foundation.

Data Analysis Transparency:

The paper mentions the use of a fuzzy logic model to interpret and contrast data, but it falls short in elucidating the specifics of this analysis. A more transparent presentation of the data analysis process, including how the fuzzy logic model contributes to insights, will enhance the paper's credibility.

Detailed Insights into Evaluation Aspects:

The three aspects highlighted for evaluation (satisfaction, ease of use, and efficiency) are crucial, but the paper lacks detailed insights into specific findings within each aspect. Providing a more granular breakdown of results within these aspects would offer a richer understanding of user perceptions and experiences.

Generalizability and Limitations:

The paper asserts a high likelihood of user recommendation and overall quality of user experience based on the results. However, it is essential to discuss the generalizability of these findings and acknowledge any limitations in the study, such as specific user demographics or contextual factors that might influence the results.

Comparative Analysis:

The paper could benefit from a more explicit comparative analysis, especially when contrasting quantitative and qualitative data. A deeper exploration of instances where these data sources converge or diverge would provide a more nuanced interpretation of the results.

Practical Implications and Design Recommendations:

While the paper highlights contributions to mobile application usability, it could be strengthened by providing practical implications derived from the study's findings. Offering specific design recommendations based on user feedback would enhance the paper's utility for practitioners and designers in the field of residential energy management applications.

Please avoid citing sources that were published before to 2019. Cite current research that is pertinent to your topic. The study also lacks sufficient citations. Another critical step is to compare the topic of the article to other relevant recent publications or works in order to widen the research's repercussions beyond the issue. Authors can use and depend on these essential works while addressing the topic of their 

Yan, Shu-Rong, et al. "Implementation of a Product-Recommender System in an IoT-Based Smart Shopping Using Fuzzy Logic and Apriori Algorithm." IEEE Transactions on Engineering Management (2022).

Martín, Juan Carlos, and Alessandro Indelicato. "Comparing a Fuzzy Hybrid Approach with Invariant MGCFA to Study National Identity." Applied Sciences 13.3 (2023): 1657.

Gardas, Bhaskar B., et al. "A fuzzy-based method for objects selection in blockchain-enabled edge-IoT platforms using a hybrid multi-criteria decision-making model." Applied Sciences 12.17 (2022): 8906.

Comments on the Quality of English Language

 Minor editing of English language required

Author Response

For research article

Response to Reviewer 3 Comments

1. Summary

Thank you very much for taking the time to review this manuscript. We extend our gratitude for your diligent review of our manuscript and for offering invaluable feedback. We appreciate your recognition of the contribution our paper makes. We're thrilled you enjoyed the article!.

2. Questions for General Evaluation

Reviewer’s Evaluation

Response and Revisions

Does the introduction provide sufficient background and include all relevant references?

Can be improved

[Please give your response if necessary. Or you can also give your corresponding response in the point-by-point response letter. The same as below]

Are all the cited references relevant to the research?

Can be improved

Is the research design appropriate?

Must be improved

Are the methods adequately described?

Can be improved

Are the results clearly presented?

Can be improved

Are the conclusions supported by the results?

Must be improved

3. Thank you for your insightful comments and suggestions on our paper. We will address the following key points in our revision:

Comments 1:  Clarity of Methodology and Integration:

The paper mentions a hybrid approach using both quantitative and qualitative methods but lacks clarity on how these methods are integrated within the user-centered design framework. Providing a more detailed explanation of the integration process and the rationale behind it would enhance the paper's methodological robustness.

Response to comments 1: We appreciate your comment about the need for greater clarity on the integration of quantitative and qualitative methods in our user-centered design framework. In the article, we added a more detailed explanation of the integration process in the methodology and the rationale behind it.

Comments 2: Fuzzy Logic Model Implementation:

While the paper introduces the use of a fuzzy logic model for interpreting and contrasting data, it lacks in-depth information on the model's design and implementation specifics. A more detailed exposition on how the fuzzy logic model was employed, including parameters and decision-making processes, is crucial for transparency and reproducibility.

Response to comments 2: Thank you for your comment. In the article, we include more specific details on how we designed and implemented the fuzzy logic model. Adding other controls was considered but would increase the complexity of the model and its subsequent analysis.

Comments 3: Metric Selection and Justification:

The paper briefly mentions the use of metrics like UEQ-S and SUS but lacks a thorough justification for their selection. Providing a rationale for choosing these specific metrics over alternatives and discussing their appropriateness for the study's objectives would strengthen the paper's methodological foundation.

Response to comments 3: Thank you for your comment. In the article, we provide a more thorough rationale for the choice of metrics such as UEQ-S and SUS, explaining why we consider them suitable for our research objectives compared to other alternatives available in the field.

Comments 4: Data Analysis Transparency:

The paper mentions the use of a fuzzy logic model to interpret and contrast data, but it falls short in elucidating the specifics of this analysis. A more transparent presentation of the data analysis process, including how the fuzzy logic model contributes to insights, will enhance the paper's credibility.

Response to comments 4: Thank you for your comment. We provide a more detailed presentation of the data analysis process, including how the fuzzy logic model contributes to the insights derived from the study.

Comments 5: Detailed Insights into Evaluation Aspects:

The three aspects highlighted for evaluation (satisfaction, ease of use, and efficiency) are crucial, but the paper lacks detailed insights into specific findings within each aspect. Providing a more granular breakdown of results within these aspects would offer a richer understanding of user perceptions and experiences.

Response to comments 5: Thank you for your feedback. We will be sure to provide a richer understanding of user perceptions and experiences by providing more specific details about these findings.

Comments 6: Generalizability and Limitations:

The paper asserts a high likelihood of user recommendation and overall quality of user experience based on the results. However, it is essential to discuss the generalizability of these findings and acknowledge any limitations in the study, such as specific user demographics or contextual factors that might influence the results.

Response to comments 6: Thank you for your comment. In the article, we more clearly address the results and detail the limitations of the study, such as user demographics and contextual factors that could influence the conclusions.

Comments 7: Comparative Analysis:

The paper could benefit from a more explicit comparative analysis, especially when contrasting quantitative and qualitative data. A deeper exploration of instances where these data sources converge or diverge would provide a more nuanced interpretation of the results.

Response to comments 7: Thank you for your comment. We provided a more in-depth exploration of how qualitative and quantitative data provides information, which allowed for a more nuanced interpretation of our results.

Comments 8: Practical Implications and Design Recommendations:

While the paper highlights contributions to mobile application usability, it could be strengthened by providing practical implications derived from the study's findings. Offering specific design recommendations based on user feedback would enhance the paper's utility for practitioners and designers in the field of residential energy management applications.

Please avoid citing sources that were published before to 2019. Cite current research that is pertinent to your topic. The study also lacks sufficient citations. Another critical step is to compare the topic of the article to other relevant recent publications or works to widen the research's repercussions beyond the issue.

Response to comments 8: Thank you for your comment. We have strengthened this section by providing specific design recommendations derived from user feedback and study results.

Thank you once more for your valuable time and professional guidance. Your insights are essential for improving the quality of our paper. We have meticulously reviewed your suggestions and implemented corresponding adjustments in the revised version. We eagerly await any further feedback on the latest iteration of the manuscript. Your continued professional support is greatly appreciated, and we anticipate making significant progress with your assistance in advancing this paper.

Sincerely,

Round 2

Reviewer 3 Report

Comments and Suggestions for Authors

Well done.